# DESPOT: Online POMDP Planning with Regularization

**Adhiraj Somani**       **Nan Ye**       **David Hsu**       **Wee Sun Lee**

Department of Computer Science
National University of Singapore
adhirajsomani@gmail.com, {yenan,dyhsu,leews}@comp.nus.edu.sg

## Abstract

POMDPs provide a principled framework for planning under uncertainty, but are computationally intractable, due to the "curse of dimensionality" and the "curse of history". This paper presents an online POMDP algorithm that alleviates these difficulties by focusing the search on a set of randomly sampled *scenarios*. A *Determinized Sparse Partially Observable Tree* (DESPOT) compactly captures the execution of all policies on these scenarios. Our *Regularized DESPOT* (R-DESPOT) algorithm searches the DESPOT for a policy, while optimally balancing the size of the policy and its estimated value obtained under the sampled scenarios. We give an output-sensitive performance bound for all policies derived from a DESPOT, and show that R-DESPOT works well if a small optimal policy exists. We also give an anytime algorithm that approximates R-DESPOT. Experiments show strong results, compared with two of the fastest online POMDP algorithms. Source code along with experimental settings are available at http://bigbird.comp.nus.edu.sg/pmwiki/farm/appl/.

## 1   Introduction

Partially observable Markov decision processes (POMDPs) provide a principled general framework for planning in partially observable stochastic environments. However, POMDP planning is computationally intractable in the worst case [11]. The challenges arise from three main sources. First, a POMDP may have a large number of states. Second, as the state is not fully observable, the agent must reason with *beliefs*, which are probability distributions over the states. Roughly, the size of the belief space grows exponentially with the number of states. Finally, the number of action-observation histories that must be considered for POMDP planning grows exponentially with the planning horizon. The first two difficulties are usually referred to as the "curse of dimensionality", and the last one, the "curse of history". To address these difficulties, online POMDP planning (see [17] for a survey) chooses one action at a time and interleaves planning and plan execution. At each time step, the agent performs a $D$-step lookahead search. It plans the immediate next action for the current belief only and reasons in the neighborhood of the current belief, rather than over the entire belief space. Our work adopts this online planning approach.

Recently an online POMDP planning algorithm called POMCP has successfully scaled up to very large POMDPs [18]. POMCP, which is based on Monte Carlo tree search, tries to break the two curses by *sampling states* from the current belief and *sampling histories* with a black-box simulator. It uses the UCT algorithm [9] to control the exploration-exploitation trade-off during the online lookahead search. However, UCT is sometimes overly greedy and suffers the worst-case performance of $\Omega(\exp(\exp(\ldots \exp(1)\ldots)))$[1] samples to find a sufficiently good action [4].

This paper presents a new algorithm for online POMDP planning. It enjoys the same strengths as POMCP—breaking the two curses through sampling—but avoids POMCP's extremely poor worst-case behavior by evaluating policies on a small number of sampled *scenarios* [13]. In each planning step, the algorithm searches for a good policy derived from a *Determinized Sparse Partially Observable Tree* (DESPOT) for the current belief, and executes the policy for one step. A DESPOT summarizes the execution of *all policies* under $K$ sampled scenarios. It is structurally similar to a standard belief tree, but contains only belief nodes reachable under the $K$ scenarios

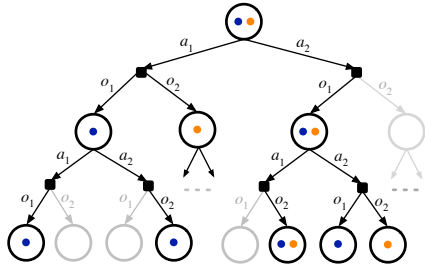

Figure 1: A belief tree of height $D = 2$ (gray) and a corresponding DESPOT (black) obtained with 2 sampled scenarios. Every tree nodes represents a belief. Every colored dot represents a scenario.

(Figure 1). We can view a DESPOT as a sparsely sampled belief tree. While a belief tree of height $D$ contains $\mathcal{O}(|A|^D|Z|^D)$ nodes, where $|A|$ and $|Z|$ are the sizes of the action set and the observation set, respectively, a corresponding DESPOT contains only $\mathcal{O}(|A|^D K)$ nodes, leading to dramatic improvement in computational efficiency when $K$ is small.

One main result of this work is an output-sensitive bound, showing that a small number of sampled scenarios is sufficient to give a good estimate of the true value of *any* policy $\pi$, provided that the size of $\pi$ is small (Section 3). Our *Regularized DESPOT* (R-DESPOT) algorithm interprets this lower bound as a regularized utility function, which it uses to optimally balance the size of a policy and its estimated performance under the sampled scenarios. We show that R-DESPOT computes a near-optimal policy whenever a small optimal policy exists (Section 4). For anytime online planning, we give a heuristic approximation, *Anytime Regularized DESPOT* (AR-DESPOT), to the R-DESPOT algorithm (Section 5). Experiments show strong results of AR-DESPOT, compared with two of the fastest online POMDP algorithms (Section 6).

## 2 Related Work

There are two main approaches to POMDP planning: offline policy computation and online search. In offline planning, the agent computes beforehand a policy contingent upon all possible future scenarios and executes the computed policy based on the observations received. Although offline planning algorithms have achieved dramatic progress in computing near-optimal policies (e.g., [15, 21, 20, 10]), they are difficult to scale up to very large POMDPs, because of the exponential number of future scenarios that must be considered.

In contrast, online planning interleaves planning and plan execution. The agent searches for a single best action for the current belief only, executes the action, and updates the belief. The process then repeats at the new belief. A recent survey [17] lists three main categories of online planning algorithms: heuristic search, branch-and-bound pruning, and Monte Carlo sampling. AR-DESPOT contains elements of all three, and the idea of constructing DESPOTs through deterministic sampling is related to those in [8, 13]. However, AR-DESPOT balances the size of a policy and its estimated performance during the online search, resulting in improved performance for suitable planning tasks.

During the online search, most algorithms, including those based on Monte Carlo sampling (e.g., [12, 1]), explicitly represents the belief as a probability distribution over the state space. This, however, limits their scalability for large state spaces, because a single belief update can take time quadratic in the number of states. In contrast, DESPOT algorithms represent the belief as a set of particles, just as POMCP [18] does, and do not perform belief update during the online search.

Online search and offline policy computation are complementary and can be combined, e.g., by using approximate or partial policies computed offline as the default policies at the bottom of the search tree for online planning (e.g., [2, 5]) or as macro-actions to shorten the search horizon [7].

## 3 Determinized Sparse Partially Observable Trees

### 3.1 POMDP Preliminaries

A POMDP is formally a tuple $(S, A, Z, T, O, R)$, where $S$ is a set of states, $A$ is a set of actions, $Z$ is a set of observations, $T(s, a, s') = p(s'|s, a)$ is the probability of transitioning to state $s'$ when the agent takes action $a$ in state $s$, $O(s, a, z) = p(z|s, a)$ is the probability of observing $z$ if the agent takes action $a$ and ends in state $s$, and $R(s, a)$ is the immediate reward for taking action $a$ in state $s$.

A POMDP agent does not know the true state, but receives observations that provide partial information on the state. The agent maintains a *belief*, often represented as a probability distribution over $S$. It starts with an initial belief $b_0$. At time $t$, it updates the belief $b_t$ according to Bayes' rule by incorporating information from the action taken at time $t-1$ and the resulting observation: $b_t = \tau(b_{t-1}, a_{t-1}, z_t)$. A policy $\pi : \mathcal{B} \mapsto A$ specifies the action $a \in A$ at belief $b \in \mathcal{B}$. The *value* of a policy $\pi$ at a belief $b$ is the expected total discounted reward obtained by following $\pi$ with initial belief $b$: $V_\pi(b) = \mathrm{E}\left(\sum_{t=0}^{\infty} \gamma^t R(s_t, \pi(b_t)) \mid b_0 = b\right)$, for some discount factor $\gamma \in [0, 1)$.

One way of online POMDP planning is to construct a belief tree (Figure 1), with the current belief $b_0$ as the initial belief at the root of the tree, and perform lookahead search on the tree for a policy $\pi$ that maximizes $V_\pi(b_0)$. Each node of the tree represents a belief. A node branches into $|A|$ action edges, and each action edge branches further into $|Z|$ observation edges. If a node and its child represent beliefs $b$ and $b'$, respectively, then $b' = \tau(b, a, z)$ for some $a \in A$ and $z \in Z$. To search a belief tree, we typically truncate it at a maximum depth $D$ and perform a post-order traversal. At each leaf node, we simulate a *default policy* to obtain a lower bound on its value. At each internal node, we apply Bellman's principle of optimality to choose a best action:

$$V(b) = \max_{a \in A} \left\{ \sum_{s \in S} b(s) R(s, a) + \gamma \sum_{z \in Z} p(z|b, a) V\big(\tau(b, a, z)\big) \right\}, \qquad (1)$$

which recursively computes the maximum value of action branches and the average value of observation branches. The results are an approximately optimal policy $\hat{\pi}$, represented as a *policy tree*, and the corresponding value $V_{\hat{\pi}}(b_0)$. A policy tree retains only the *chosen* action branches, but *all* observation branches from the belief tree[2]. The *size* of such a policy is the number of tree nodes.

Our algorithms represent a belief as a set of *particles*, i.e., sampled states. We start with an initial belief. At each time step, we search for a policy $\hat{\pi}$, as described above. The agent executes the first action $a$ of $\hat{\pi}$ and receives a new observation $z$. We then apply particle filtering to incorporate information from $a$ and $z$ into an updated new belief. The process then repeats.

### 3.2 DESPOT

While a standard belief tree captures the execution of all policies under all possible scenarios, a DESPOT captures the execution of all policies under a set of sampled scenarios (Figure 1). It contains all the action branches, but only the observation branches under the sampled scenarios.

We define DESPOT constructively by applying a *deterministic simulative model* to all possible action sequences under $K$ scenarios sampled from an initial belief $b_0$. A scenario is an abstract simulation trajectory starting with some state $s_0$. Formally, a *scenario* for a belief $b$ is a random sequence $\phi = (s_0, \phi_1, \phi_2, \ldots)$, in which the start state $s_0$ is sampled according to $b$ and each $\phi_i$ is a real number sampled independently and uniformly from the range $[0, 1]$. The deterministic simulative model is a function $g: S \times A \times \mathrm{R} \mapsto S \times Z$, such that if a random number $\phi$ is distributed uniformly over $[0, 1]$, then $(s', z') = g(s, a, \phi)$ is distributed according to $p(s', z'|s, a) = T(s, a, s')O(s', a, z')$. When we simulate this model for an action sequence $(a_1, a_2, a_3, \ldots)$ under a scenario $(s_0, \phi_1, \phi_2, \ldots)$, the simulation generates a trajectory $(s_0, a_1, s_1, z_1, a_2, s_2, z_2, \ldots)$, where $(s_t, z_t) = g(s_{t-1}, a_t, \phi_t)$ for $t = 1, 2, \ldots$. The simulation trajectory traces out a path $(a_1, z_1, a_2, z_2, \ldots)$ from the root of the standard belief tree. We add all the nodes and edges on this path to the DESPOT. Each DESPOT node $b$ contains a set $\Phi_b$, consisting of all scenarios that it encounters. The start states of the scenarios in $\Phi_b$ form a particle set that represents $b$ approximately. We insert the scenario $(s_0, \phi_0, \phi_1, \ldots)$ into the set $\Phi_{b_0}$ and insert $(s_t, \phi_{t+1}, \phi_{t+2}, \ldots)$ into the set $\Phi_{b_t}$ for the belief node $b_t$ reached at the end of the subpath $(a_1, z_1, a_2, z_2, \ldots, a_t, z_t)$, for $t = 1, 2, \ldots$. Repeating this process for every action sequence under every sampled scenario completes the construction of the DESPOT.

A DESPOT is determined completely by the $K$ scenarios, which are sampled randomly a priori. Intuitively, a DESPOT is a standard belief tree with some observation branches removed. While a belief tree of height $D$ has $\mathcal{O}(|A|^D |Z|^D)$ nodes, a corresponding DESPOT has only $\mathcal{O}(|A|^D K)$ nodes, because of reduced observation branching under the sampled scenarios. Hence the name *Determinized Sparse Partially Observable Tree* (DESPOT).

To evaluate a policy $\pi$ under sampled scenarios, define $V_{\pi,\phi}$ as the total discounted reward of the trajectory obtained by simulating $\pi$ under a scenario $\phi$. Then $\hat{V}_\pi(b) = \sum_{\phi \in \Phi_b} V_{\pi,\phi} / |\Phi_b|$ is an estimate of $V_\pi(b)$, the value of $\pi$ at $b$, under a set of scenarios, $\Phi_b$. We then apply the usual belief tree search from the previous subsection to a DESPOT to find a policy having good performance under the sampled scenarios. We call this algorithm *Basic DESPOT* (B-DESPOT).

The idea of using sampled scenarios for planning is exploited in hindsight optimization (HO) as well [3, 22]. HO plans for each scenario independently and builds $K$ separate trees, each with $\mathcal{O}(|A|^D)$ nodes. In contrast, DESPOT captures all $K$ scenarios in a single tree with $\mathcal{O}(|A|^D K)$ nodes and allows us to reason with all scenarios simultaneously. For this reason, DESPOT can provide stronger performance guarantees than HO.

# 4   Regularized DESPOT

To search a DESPOT for a near-optimal policy, B-DESPOT chooses a best action at every internal node of the DESPOT, according to the scenarios it encounters. This, however, may cause *overfitting*: the chosen policy optimizes for the sampled scenarios, but does not perform well in general, as many scenarios are not sampled. To reduce overfitting, our R-DESPOT algorithm leverages the idea of *regularization*, which balances the estimated performance of a policy under the sampled scenarios and the policy size. If the subtree at a DESPOT node is too large, then the performance of a policy for this subtree may not be estimated reliably with $K$ scenarios. Instead of searching the subtree for a policy, R-DESPOT terminates the search and uses a simple default policy from this node onwards.

To derive R-DESPOT, we start with two theoretical results. The first one provides an output-sensitive lower bound on the performance of any arbitrary policy derived from a DESPOT. It implies that despite its sparsity, a DESPOT contains sufficient information for approximate policy evaluation, and the accuracy depends on the size of the policy. The second result shows that by optimizing this bound, we can find a policy with small size and high value. For convenience, we assume that $R(s, a) \in [0, R_{\max}]$ for all $s \in S$ and $a \in A$, but the results can be easily extended to accommodate negative rewards. The proofs of both results are available in the supplementary material.

Formally, a policy tree *derived* from a DESPOT contains the same root as the DESPOT, but only one action branch at each internal node. Let $\Pi_{b_0, D, K}$ denote the class of all policy trees derived from DESPOTs that have height $D$ and are constructed from $K$ sampled scenarios for belief $b_0$. Like a DESPOT, a policy tree $\pi \in \Pi_{b_0, D, K}$ may not contain all observation branches. If the execution of $\pi$ encounters an observation branch not present in $\pi$, we simply follow the default policy from then on. Similarly, we follow the default policy, when reaching a leaf node. We now bound the error on the estimated value of a policy derived from a DESPOT.

**Theorem 1** *For any $\tau, \alpha \in (0, 1)$, every policy tree $\pi \in \Pi_{b_0, D, K}$ satisfies*

$$V_\pi(b_0) \geq \frac{1-\alpha}{1+\alpha} \hat{V}_\pi(b_0) - \frac{R_{\max}}{(1+\alpha)(1-\gamma)} \cdot \frac{\ln(4/\tau) + |\pi| \ln(KD|A||Z|)}{\alpha K}, \qquad (2)$$

*with probability at least $1 - \tau$, where $\hat{V}_\pi(b_0)$ is the estimated value of $\pi$ under any set of $K$ randomly sampled scenarios for belief $b_0$.*

The second term on the right hand side (RHS) of (2) captures the additive error in estimating the value of policy tree $\pi$, and depends on the size of $\pi$. We can make this error arbitrarily small by choosing a suitably large $K$, the number of sampled scenarios. Furthermore, this error grows logarithmically with $|A|$ and $|Z|$, indicating that the approximation scales well with large action and observation sets. The constant $\alpha$ can be tuned to tighten the bound. A smaller $\alpha$ value allows the first term on the RHS of (2) to approximate $\hat{V}_\pi$ better, but increases the additive error in the second term. We have specifically constructed the bound in this multiplicative-additive form, due to Haussler [6], in order to apply efficient dynamic programming techniques in R-DESPOT.

Now a natural idea is to search for a near-optimal policy $\pi$ by maximizing the RHS of (2), which guarantees the performance of $\pi$ by accounting for both the estimated performance and the size of $\pi$.

**Theorem 2** *Let $\pi^*$ be an optimal policy at a belief $b_0$. Let $\pi$ be a policy derived from a DESPOT that has height $D$ and is constructed from $K$ randomly sampled scenarios for belief $b_0$. For any $\tau, \alpha \in (0, 1)$, if $\pi$ maximizes*

$$\frac{1-\alpha}{1+\alpha} \hat{V}_\pi(b_0) - \frac{R_{\max}}{(1+\alpha)(1-\gamma)} \cdot \frac{|\pi| \ln(KD|A||Z|)}{\alpha K} \qquad (3)$$

*among all policies derived from the DESPOT, then*

$$V_\pi(b_0) \geq \frac{1-\alpha}{1+\alpha} V_{\pi^*}(b_0) - \frac{R_{\max}}{(1+\alpha)(1-\gamma)} \left( \frac{\ln(8/\tau) + |\pi^*| \ln(KD|A||Z|)}{\alpha K} + (1-\alpha) \left( \sqrt{\frac{2\ln(2/\tau)}{K}} + \gamma^D \right) \right),$$

*with probability at least $1 - \tau$.*

Theorem 2 implies that if a small optimal policy tree $\pi^*$ exists, then we can find a near-optimal policy with high probability by maximizing (3). Note that $\pi^*$ is a globally optimal policy at $b_0$. It may or may not lie in $\Pi_{b_0, D, K}$. The expression in (3) can be rewritten in the form $\hat{V}_\pi(b_0) - \lambda |\pi|$, similar to that of regularized utility functions in many machine learning algorithms.

We now describe R-DESPOT, which consists of two main steps. First, R-DESPOT constructs a DESPOT $T$ of height $D$ using $K$ scenarios, just as B-DESPOT does. To improve online planning performance, it may use offline learning to optimize the values for $D$ and $K$. Second, R-DESPOT performs bottom-up dynamic programming on $T$ and derive a policy tree that maximizes (3).

For a given policy tree $\pi$ derived the DESPOT $T$, we define the *regularized weighted discounted utility* (RWDU) for a node $b$ of $\pi$:

$$\nu(b) = \frac{|\mathbf{\Phi}_b|}{K} \gamma^{\Delta(b)} \hat{V}_{\pi_b}(b) - \lambda |\pi_b|,$$

where $|\mathbf{\Phi}_b|$ is the number of scenarios passing through node $b$, $\gamma$ is the discount factor, $\Delta(b)$ is the depth of $b$ in the tree $\pi$, $\pi_b$ is the subtree of $\pi$ rooted at $b$, and $\lambda$ is a fixed constant. Then the regularized utility $\hat{V}_\pi(b_0) - \lambda |\pi|$ is simply $\nu(b_0)$. We can compute $\nu(\pi_b)$ recursively:

$$\nu(b) = \hat{R}(b, a_b) + \sum_{b' \in \mathrm{CH}_\pi(b)} \nu(b') \quad \text{and}$$

$$\hat{R}(b, a_b) = \frac{1}{K} \sum_{\phi \in \mathbf{\Phi}_b} \gamma^{\Delta(b)} R(s_\phi, a_b) - \lambda.$$

where $a_b$ is the chosen action of $\pi$ at the node $b$, $\mathrm{CH}_\pi(b)$ is the set of child nodes of $b$ in $\pi$, and $s_\phi$ is the start state associated with the scenario $\phi$.

We now describe the dynamic programming procedure that searches for an optimal policy in $T$. For any belief node $b$ in $T$, let $\nu^*(b)$ be the maximum RWDU of $b$ under any policy tree $\pi$ derived from $T$. We compute $\nu^*(b)$ recursively. If $b$ is a leaf node of $T$, $\nu^*(b) = \frac{|\mathbf{\Phi}_b|}{K} \gamma^{\Delta(b)} \hat{V}_{\pi_0}(b) - \lambda$, for some default policy $\pi_0$. Otherwise,

$$\nu^*(b) = \max \left\{ \frac{|\mathbf{\Phi}_b|}{K} \gamma^{\Delta(b)} \hat{V}_{\pi_0}(b) - \lambda, \ \max_a \left\{ \hat{R}(b, a) + \sum_{b' \in \mathrm{CH}(b,a)} \nu^*(b') \right\} \right\}, \qquad (4)$$

where $\mathrm{CH}(b, a)$ is the set of child nodes of $b$ under the action branch $a$. The first maximization in (4) chooses between executing the default policy or expanding the subtree at $b$. The second maximization chooses among the different actions available. The value of an optimal policy for the DESPOT $T$ rooted at the belief $b_0$ is then $\nu^*(b_0)$ and can be computed with bottom-up dynamic programming in time linear in the size of $T$.

## 5 Anytime Regularized DESPOT

To further improve online planning performance for large-scale POMDPs, we introduce AR-DESPOT, an anytime approximation of R-DESPOT. AR-DESPOT applies heuristic search and branch-and-bound pruning to uncover the more promising parts of a DESPOT and then searches the partially constructed DESPOT for a policy that maximizes the regularized utility in Theorem 2. A brief summary of AR-DESPOT is given in Algorithm 1. Below we provides some details on how AR-DESPOT performs the heuristic search (Section 5.1) and constructs the upper and lower bounds for branch-and-bound pruning (Sections 5.2 and 5.3 ).

### 5.1 DESPOT Construction by Forward Search

AR-DESPOT incrementally constructs a DESPOT $T$ using heuristic forward search [19, 10]. Initially, $T$ contains only the root node with associated belief $b_0$ and a set $\mathbf{\Phi}_{b_0}$ of scenarios sampled according $b_0$. We then make a series of *trials*, each of which augments $T$ by tracing a path from the root to a leaf of $T$ and adding new nodes to $T$ at the end of the path. For every belief node $b$ in $T$, we maintain an upper bound $U(b)$ and a lower bound $L(b)$ on $\hat{V}_{\pi^*}(b)$, which is the value of the optimal policy $\pi^*$ for $b$ under the set of scenarios $\mathbf{\Phi}_b$. Similarly we maintain bounds $U(b, a)$ and $L(b, a)$ on the Q-value $Q_{\pi^*}(b, a) = \frac{1}{|\mathbf{\Phi}_b|} \sum_{\phi \in \mathbf{\Phi}_b} R(s_\phi, a) + \gamma \sum_{b' \in \mathrm{CH}(b,a)} \frac{|\mathbf{\Phi}_{b'}|}{|\mathbf{\Phi}_b|} \hat{V}_{\pi^*}(b')$. A trial starts the root of $T$. In each step, it chooses the action branch $a^*$ that maximizes $U(b, a)$ for the current node $b$ and then chooses the observation branch $z^*$ that maximizes the *weighted excess uncertainty* at the child node $b' = \tau(b, a^*, z)$:

$$\mathrm{WEU}(b') = \frac{|\mathbf{\Phi}_{b'}|}{|\mathbf{\Phi}_b|} \mathrm{excess}(b'),$$

where $\mathrm{excess}(b') = U(b') - L(b') - \epsilon \gamma^{-\Delta(b')}$ [19] and $\epsilon$ is a constant specifying the desired gap between the upper and lower bounds at the root $b_0$. If the chosen node $\tau(b, a^*, z^*)$ has negative

**Algorithm 1** AR-DESPOT

1: Set $b_0$ to the initial belief.
2: **loop**
3:    $T \leftarrow$ BUILDDESPOT($b_0$).
4:    Compute an optimal policy $\pi^*$ for $T$ using (4)
5:    Execute the first action of $a$ of $\pi^*$.
6:    Receive observation $z$.
7:    Update the belief $b_0 \leftarrow \tau(b_0, a, z)$.

BUILDDESPOT($b_0$)

1: Sample a set $\mathbf{\Phi}_{b_0}$ of $K$ random scenarios for $b_0$.
2: Insert $b_0$ into $T$ as the root node.
3: **while** time permitting **do**
4:    $b \leftarrow$ RUNTRIAL($b_0, T$).
5:    Back up upper and lower bounds for every node on the path from $b$ to $b_0$.
6: **return** $T$

RUNTRIAL($b, T$)

1: **if** $\Delta(b) > D$ **then**
2:    **return** $b$
3: **if** $b$ is a leaf node **then**
4:    Expand $b$ one level deeper, and insert all new nodes into $T$ as children of $b$.
5: $a^* \leftarrow \arg\max_{a \in A} U(b, a)$.
6: $z^* \leftarrow \arg\max_{z \in Z_{b,a^*}} \text{WEU}(\tau(b, a^*, z))$.
7: $b \leftarrow \tau(b, a^*, z^*)$.
8: **if** $\text{WEU}(b) \geq 0$ **then**
9:    **return** RUNTRIAL($b, T$)
10: **else**
11:    **return** $b$

excess uncertainty, the trial ends. Otherwise it continues until reaching a leaf node of $T$. We then expand the leaf node $b$ one level deeper by adding new belief nodes for every action and every observation as children of $b$. Finally we trace the path backward to the root and perform backup on both the upper and lower bounds at each node along the way. For the lower-bound backup,

$$L(b) = \max_{a \in A} \left\{ \frac{1}{|\mathbf{\Phi}_b|} \sum_{\boldsymbol{\phi} \in \mathbf{\Phi}_b} R(s_{\boldsymbol{\phi}}, a) + \gamma \sum_{z \in Z_{b,a}} \frac{|\mathbf{\Phi}_{\tau(b,a,z)}|}{|\mathbf{\Phi}_b|} L(\tau(b, a, z)) \right\}. \tag{5}$$

where $Z_{b,a}$ is the set of observations encountered when action $a$ is taken at $b$ under all scenarios in $\mathbf{\Phi}_b$. The upper bound backup is the same. We repeat the trials as long as time permits, thus making the algorithm anytime.

## 5.2 Initial Upper Bounds

There are several approaches for constructing the initial upper bound at a node $b$ of a DESPOT. A simple one is the uninformative bound of $R_{\max}/(1 - \gamma)$. To obtain a tighter bound, we may exploit domain-specific knowledge. Here we give a domain-independent construction, which is the average upper bound over all scenarios in $\mathbf{\Phi}_b$. The upper bound for a particular scenario $\boldsymbol{\phi} \in \mathbf{\Phi}_b$ is the maximum value achieved by any arbitrary policy under $\boldsymbol{\phi}$. Given $\boldsymbol{\phi}$, we have a deterministic planning problem and solve it by dynamic programming on a trellis of $D$ time slices. Trellis nodes represent states, and edges represent actions at each time step. The path with highest value in the trellis gives the upper bound under $\boldsymbol{\phi}$. Repeating this procedure for every $\boldsymbol{\phi} \in \mathbf{\Phi}_b$ and taking the average gives an upper bound on the value of $b$ under the set $\mathbf{\Phi}_b$. It can be computed in $\mathcal{O}(K|S||A|D)$ time.

## 5.3 Initial Lower Bounds and Default Policies

To construct the lower bound at a node $b$, we may simulate any policy for $N$ steps under the scenarios in $\mathbf{\Phi}_b$ and compute the average total discounted reward, all in $\mathcal{O}(|\mathbf{\Phi}_b|N)$ time. One possibility is to use a fixed-action policy for this purpose. A better one is to handcraft a policy that chooses an action based on the history of actions and observations, a technique used in [18]. However, it is often difficult to handcraft effective history-based policies. We thus construct a policy using the belief $b$: $\pi(b) = f(\Lambda(b))$, where $\Lambda(b)$ is the mode of the probability distribution $b$ and $f: S \rightarrow A$ is a mapping that specifies the action at the state $s \in S$. It is much more intuitive to construct $f$, and we can approximate $\Lambda(b)$ easily by determining the most frequent state using $\mathbf{\Phi}_b$. Note that although history-based policies satisfy the requirements of Theorem 1, belief-based policies do not. The difference is, however, unlikely to be significant to affect performance in practice.

# 6 Experiments

To evaluate AB-DESPOT experimentally, we compared it with four other algorithms. Anytime Basic DESPOT (AB-DESPOT) is AR-DESPOT without the dynamic programming step that computes RWDU. It helps to understand the benefit of regularization. AEMS2 is an early successful online POMDP algorithm [16, 17]. POMCP has scaled up to very large POMDPs [18]. SARSOP is a state-of-the-art offline POMDP algorithm [10]. It helps to calibrate the best performance achievable for POMDPs of moderate size. In our online planning tests, each algorithm was given exactly 1 second per step to choose an action. For AR-DESPOT and AB-DESPOT, $K = 500$ and $D = 90$. The regularization parameter $\lambda$ for AR-DESPOT was selected offline by running the algorithm with a training set distinct from the online test set. The discount factor is $\gamma = 0.95$. For POMCP, we used the implementation from the original authors[3], but modified it in order to support very large number of observations and strictly follow the 1-second time limit for online planning.

We evaluated the algorithms on four domains, including a very large one with about $10^{56}$ states (Table 1). In summary, compared with AEMS2, AR-DESPOT is competitive on smaller POMDPs, but scales up much better on large POMDPs. Compared with POMCP, AR-DESPOT performs better than POMCP on the smaller POMDPs and scales up just as well.

We first tested the algorithms on *Tag* [15], a standard benchmark problem. In Tag, the agent's goal is to find and tag a target that intentionally moves away. Both the agent and target operate in a grid with 29 possible positions. The agent knows its own position but can observe the target's position only if they are in the same location. The agent can either stay in the same position or move to the four adjacent positions, paying a cost for each move. It can also perform the tag action and is rewarded if it successfully tags the target, but is penalized if it fails. For POMCP, we used the Tag implementation that comes with the package, but modified it slightly to improve its default rollout policy. The modified policy always tags when the agent is in the same position as the robot, providing better performance. For AR-DESPOT, we use a simple particle set default policy, which moves the agent towards the mode of the target in the particle set. For the upper bound, we average the upper bound for each particle as described in Section 5.2. The results (Table 1) show that AR-DESPOT gives comparable performance to AEMS2.

Theorem 1 suggests that AR-DESPOT may still perform well when the observation space is large, if a good small policy exists. To examine the performance of AR-DESPOT on large observation spaces, we experimented with an augmented version of Tag called *LaserTag*. In LaserTag, the agent moves in a $7 \times 11$ rectangular grid with obstacles placed in 8 random cells. The behavior of the agent and opponent are identical to that in Tag, except that in LaserTag the agent knows it location before the game starts, whereas in Tag this happens only after the first observation is seen.

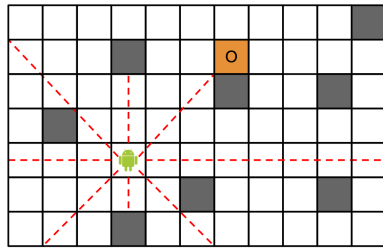

The agent is equipped with a laser that gives distance estimates in 8 directions. The distance between 2 adjacent cells is considered one unit, and the laser reading in each direction is generated from a normal distribution centered at the true distance of the agent from the nearest obstacle in that direction, with a standard deviation of 2.5 units. The readings are discretized into whole units, so an observation comprises a set of 8 integers. For a map of size $7 \times 11$, $|Z|$ is of the order of $10^6$. The environment for LaserTag is shown in Figure 2. As can be seen from Table 1, AR-DESPOT outperforms POMCP on this problem. We can also see the effect of regularization by comparing AR-DESPOT with AB-DESPOT. It is not feasible to run AEMS2 or SARSOP on this problem in reasonable time because of the very large observation space.

Figure 2: Laser Tag. The agent moves in a $7 \times 11$ grid with obstacles placed randomly in 8 cells. It is equipped with a noisy laser that gives distance estimates in 8 directions.

To demonstrate the performance of AR-DESPOT on large state spaces, we experimented with the *RockSample* problem [19]. The *RockSample*$(n, k)$ problem mimics a robot moving in an $n \times n$ grid containing $k$ rocks, each of which may be good or bad. At each step, the robot either moves to an adjacent cell, samples a rock, or senses a rock. Sampling gives a reward of +10 if the rock is good and -10 otherwise. Both moving and sampling produce a null observation. Sensing produces an observation in $\{good, bad\}$, with the probability of producing the correct observation decreasing

Table 1: Performance comparison, according to the average total discounted reward achieved. The results for SARSOP and AEMS2 are replicated from [14] and [17], respectively. SARSOP and AEMS2 failed to run on some domains, because their state space or observation space is too large. For POMCP, both results from our own tests and those from [18] (in parentheses) are reported. We could not reproduce the earlier published results, possibly because of the code modification and machine differences.

| | Tag | LaserTag | RS(7,8) | RS(11,11) | RS(15,15) | Pocman |
|---|---|---|---|---|---|---|
| No. States $|S|$ | 870 | 4,830 | 12,544 | 247,808 | 7,372,800 | $\sim 10^{56}$ |
| No. Actions $|A|$ | 5 | 5 | 13 | 16 | 20 | 4 |
| No. Observations $|Z|$ | 30 | $\sim 1.5 \times 10^6$ | 3 | 3 | 3 | 1024 |
| SARSOP | $-6.03 \pm 0.12$ | – | $21.47 \pm 0.04$ | $21.56 \pm 0.11$ | – | – |
| AEMS2 | $-6.19 \pm 0.15$ | – | $21.37 \pm 0.22$ | – | – | – |
| POMCP | $-7.14 \pm 0.28$ | $-19.58 \pm 0.06$ | $16.80 \pm 0.30$ | $18.10 \pm 0.36$ | $12.23 \pm 0.32$ | $294.16 \pm 4.06$ |
| | | | $(20.71 \pm 0.21)$ | $(20.01 \pm 0.23)$ | $(15.32 \pm 0.28)$ | |
| AB-DESPOT | $-6.57 \pm 0.26$ | $-11.13 \pm 0.30$ | $21.07 \pm 0.32$ | $21.60 \pm 0.32$ | $18.18 \pm 0.30$ | $290.34 \pm 4.12$ |
| AR-DESPOT | $-6.26 \pm 0.28$ | $-9.34 \pm 0.26$ | $21.08 \pm 0.30$ | $21.65 \pm 0.32$ | $18.57 \pm 0.30$ | $307.96 \pm 4.22$ |

exponentially with the agent's distance from the rock. A terminal state is reached when the agent moves past the east edge of the map. For AR-DESPOT, we use a default policy derived from the particle set as follows: a new state is created with the positions of the robot and the rocks unchanged, and each rock is labeled as good or bad depending on whichever condition is more prevalent in the particle set. The optimal policy for the resulting state is used as the default policy. The optimal policy for all states is computed before the algorithm begins, using dynamic programming with the same horizon length as the maximum depth of the search tree. For the initial upper bound, we use the method described in Section 5.2. As in [18], we use a particle filter to represent the belief to examine the behavior of the algorithms in very large state spaces. For POMCP, we used the implementation in [18] but ran it on the same platform as AR-DESPOT. As the results for our runs of POMCP are poorer than those reported in [18], we also reproduce their reported results in Table 1. The results in Table 1 indicate that AR-DESPOT is able to scale up to very large state spaces. Regularization does not appear beneficial to this problem, possibly because it is mostly deterministic, except for the sensing action.

Finally, we implemented *Pocman*, the partially observable version of the video game *Pacman*, as described in [18]. *Pocman* has an extremely large state space of approximately $10^{56}$. We compute an approximate upper bound for a belief by summing the following quantities for each particle in it, and taking the average over all particles: reward for eating each pellet discounted by its distance from pocman; reward for clearing the level discounted by the maximum distance to a pellet; default per-step reward of $-1$ for a number of steps equal to the maximum distance to a pellet; penalty for eating a ghost discounted by the distance to the closest ghost being chased (if any); penalty for dying discounted by the average distance to the ghosts; and half the penalty for hitting a wall if pocman tries to double back along its direction of movement. This need not always be an upper bound, but AR-DESPOT can be modified to run even when this is the case. For the lower bound, we use a history-based policy that chases a random ghost, if visible, when pocman is under the effect of a powerpill, and avoids ghosts and doubling-back when it is not. This example shows that AR-DESPOT can be used successfully even in cases of extremely large state space.

# 7 Conclusion

This paper presents DESPOT, a new approach to online POMDP planning. Our R-DESPOT algorithm and its anytime approximation, AR-DESPOT, search a DESPOT for an approximately optimal policy, while balancing the size of the policy and the accuracy on its value estimate. Theoretical analysis and experiments show that the new approach outperforms two of the fastest online POMDP planning algorithms. It scales up better than AEMS2, and it does not suffer the extremely poor worst-case behavior of POMCP. The performance of AR-DESPOT depends on the upper and lower bounds supplied. Effective methods for automatic construction of such bounds will be an interesting topic for further investigation.

**Acknowledgments.** This work is supported in part by MoE AcRF grant 2010-T2-2-071, National Research Foundation Singapore through the SMART IRG program, and US Air Force Research Laboratory under agreement FA2386-12-1-4031.

## Footnotes

[1]Composition of $D - 1$ exponential functions.

[2]A policy tree can be represented more compactly by labeling each node by the action edge that follows and then removing the action edge. We do not use this representation here.

[3]http://www0.cs.ucl.ac.uk/staff/D.Silver/web/Applications.html

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
