[Supplementary Material · supp.pdf]

# DESPOT: Online POMDP Planning with Regularization
## Supplementary Material

**Adhiraj Somani**     **Nan Ye**     **David Hsu**     **Wee Sun Lee**
Department of Computer Science, National University of Singapore
adhirajsomani@gmail.com, {yenan,dyhsu,leews}@comp.nus.edu.sg

## 1   Proof of Theorem 1

We will need two lemmas for proving Theorem 1. The first one is Haussler's bound given in [1, p. 103] (Lemma 9, part (2)).

**Lemma 1** (Haussler's bound) *Let $Z_1, \ldots, Z_n$ be i.i.d random variables with range $0 \leq Z_i \leq M$, $\mathbb{E}(Z_i) = \mu$, and $\hat{\mu} = \frac{1}{n} \sum_{i=1}^{n} Z_i$, $1 \leq i \leq n$. Assume $\nu > 0$ and $0 < \alpha < 1$. Then*

$$\Pr\left(d_\nu(\hat{\mu}, \mu) > \alpha\right) < 2e^{-\alpha^2 \nu n / M}$$

*where $d_\nu(r, s) = \frac{|r-s|}{\nu + r + s}$. As a consequence,*

$$\Pr\left(\mu < \frac{1-\alpha}{1+\alpha}\hat{\mu} - \frac{\alpha}{1+\alpha}\nu\right) < 2e^{-\alpha^2 \nu n / M}.$$

Let $\Pi_i$ be the class of policy trees in $\Pi_{b_0, D, K}$ and having size $i$. The next lemma bounds the size of $\Pi_i$.

**Lemma 2** $|\Pi_i| \leq i^{(i-2)}(|A||Z|)^i$.

**Proof.** Let $\Pi_i'$ be the class of rooted ordered trees of size $i$. $|\Pi_i'|$ is not more than the number of all trees with $i$ labeled nodes, because the in-order labeling of a tree in $\Pi_i'$ corresponds to a labeled tree. By Cayley's formula [3], the number of trees with $i$ labeled nodes is $i^{(i-2)}$, thus $|\Pi_i'| \leq i^{(i-2)}$. Recall the definition of a policy derivable from a DESPOT in Section 4 in the main text. A policy tree in $\Pi_i$ is obtained from a tree in $\Pi_i'$ by assigning the default policy to each leaf node, one of the $|A|$ possible action labels to all other nodes, and one of at most $|Z|$ possible labels to each edge. Therefore

$$|\Pi_i| \quad \leq \quad i^{(i-2)} \cdot |A|^i \cdot |Z|^{(i-1)} \quad \leq \quad i^{(i-2)}(|A||Z|)^i.$$

$\square$

In the following, we often abbreviate $V_\pi(b_0)$ and $\hat{V}_\pi(b_0)$ as $V_\pi$ and $\hat{V}_\pi$ respectively, since we will only consider the true and empirical values for a fixed but arbitrary $b_0$. Our proof follows a line of reasoning similar to [2].

**Theorem 1** *For any $\tau, \alpha \in (0, 1)$ and any set $\mathbf{\Phi}_{b_0}$ of $K$ randomly sampled scenarios for belief $b_0$, every policy tree $\pi \in \Pi_{b_0, D, K}$ satisfies*

$$V_\pi(b_0) \geq \frac{1-\alpha}{1+\alpha}\hat{V}_\pi(b_0) - \frac{R_{\max}}{(1+\alpha)(1-\gamma)} \cdot \frac{\ln(4/\tau) + |\pi|\ln\left(KD|A||Z|\right)}{\alpha K}.$$

*with probability at least $1 - \tau$, where $\hat{V}_\pi(b_0)$ denotes the estimated value of $\pi$ under $\mathbf{\Phi}_{b_0}$.*

**Proof.** Consider an arbitrary policy tree $\pi \in \Pi_{b_0,D,K}$. We know that for a random scenario $\phi$ for the belief $b_0$, executing the policy $\pi$ w.r.t. $\phi$ gives us a sequence of states and observations distributed according to the distributions $P(s'|s,a)$ and $P(z|s,a)$. Therefore, for $\pi$, its true value $V_\pi$ equals $\mathbb{E}(V_{\pi,\phi})$, where the expectation is over the distribution of scenarios. On the other hand, since $\hat{V}_\pi = \frac{1}{K}\sum_{k=1}^{K} V_{\pi,\phi_k}$, and the scenarios $\phi_0, \phi_1, \ldots, \phi_K$ are independently sampled, Lemma 1 gives

$$\Pr\left(V_\pi < \frac{1-\alpha}{1+\alpha}\hat{V}_\pi - \frac{\alpha}{1+\alpha}\epsilon_{|\pi|}\right) < 2e^{-\alpha^2 \epsilon_{|\pi|} K/M} \tag{1}$$

where $M = R_{\max}/(1-\gamma)$, and $\epsilon_i$ is chosen such that

$$2e^{-\alpha^2 \epsilon_{|\pi|} K/M} = \tau/(2i^2|\Pi_i|). \tag{2}$$

By the union bound, we have

$$\Pr\left(\exists \pi \in \Pi_{b_0,D,K}\left[V_\pi < \frac{1-\alpha}{1+\alpha}\hat{V}_\pi - \frac{\alpha}{1+\alpha}\epsilon_{|\pi|}\right]\right) \leq \sum_{i=1}^{\infty}\sum_{\pi \in \Pi_i} \Pr\left(V_\pi < \frac{1-\alpha}{1+\alpha}\hat{V}_\pi - \frac{\alpha}{1+\alpha}\epsilon_{|\pi|}\right).$$

By the choice of $\epsilon_i$'s and Inequality (1), the right hand side of the above inequality is bounded by $\sum_{i=1}^{\infty}|\Pi_i| \cdot [\tau/(2i^2|\Pi_i|)] = \pi^2\tau/12 < \tau$, where the well-known identity $\sum_{i=1}^{\infty} 1/i^2 = \pi^2/6$ is used. Hence,

$$\Pr\left(\exists \pi \in \Pi_{b_0,D,K}\left[V_\pi < \frac{1-\alpha}{1+\alpha}\hat{V}_\pi - \frac{\alpha}{1+\alpha}\epsilon_{|\pi|}\right]\right) < \tau. \tag{3}$$

Equivalently, with probability $1 - \tau$, every $\pi \in \Pi_{b_0,D,K}$ satisfies

$$V_\pi \geq \frac{1-\alpha}{1+\alpha}\hat{V}_\pi - \frac{\alpha}{1+\alpha}\epsilon_{|\pi|}. \tag{4}$$

To complete the proof, we now give an upper bound on $\epsilon_{|\pi|}$. From Equation 2, we can solve for $\epsilon_{|\pi|}$ to get $\epsilon_i = \frac{R_{\max}}{\alpha(1-\gamma)} \cdot \frac{\ln(4/\tau)+\ln(i^2|\Pi_i|)}{\alpha K}$. For any $\pi$ in $\Pi_{b_0,D,K}$, its size is at most $KD$, and $i^2|\Pi_i| \leq (i|A||Z|)^i \leq (KD|A||Z|)^i$ by Lemma 2. Thus we have

$$\epsilon_{|\pi|} \leq \frac{R_{\max}}{\alpha(1-\gamma)} \cdot \frac{\ln(4/\tau)+|\pi|\ln(KD|A||Z|)}{\alpha K}.$$

Combining this with Inequality (4), we get

$$V_\pi \geq \frac{1-\alpha}{1+\alpha}\hat{V}_\pi - \frac{R_{\max}}{(1+\alpha)(1-\gamma)} \cdot \frac{\ln(4/\tau)+|\pi|\ln(KD|A||Z|)}{\alpha K}.$$

This completes the proof. $\square$

## 2 Proof of Theorem 2

We need the following lemma for proving Theorem 2.

**Lemma 3** *For a fixed policy $\pi$ and any $\tau \in (0,1)$, with probability at least $1 - \tau$.*

$$\hat{V}_\pi \geq V_\pi - \frac{R_{\max}}{1-\gamma}\sqrt{\frac{2\ln(1/\tau)}{K}}$$

**Proof.** Let $\pi$ be a policy and $V_\pi$ and $\hat{V}_\pi$ as mentioned. Hoeffding's inequality [4] gives us

$$\Pr\left(\hat{V}_\pi \geq V_\pi - \epsilon\right) \geq 1 - e^{-K\epsilon^2/(2M^2)}$$

Let $\tau = e^{-K\epsilon^2/(2M^2)}$ and solve for $\epsilon$, then we get

$$\Pr\left(\hat{V}_\pi \geq V_\pi - \frac{R_{\max}}{1-\gamma}\sqrt{\frac{2\ln(1/\tau)}{K}}\right) \geq 1 - \tau.$$

□

**Theorem 2** *Let $\pi^*$ be an optimal policy at a belief $b_0$. Let $\pi$ be a policy derived from a DESPOT that has height $D$ and are constructed from $K$ randomly sampled scenarios for belief $b_0$. For any $\tau, \alpha \in (0,1)$, if $\pi$ maximizes*

$$\frac{1-\alpha}{1+\alpha}\hat{V}_\pi(b_0) - \frac{R_{\max}}{(1+\alpha)(1-\gamma)} \cdot \frac{|\pi|\ln(KD|A||Z|)}{\alpha K}, \tag{5}$$

*among all policies derived from the DESPOT, then*

$$V_\pi(b_0) \geq \frac{1-\alpha}{1+\alpha}V_{\pi^*}(b_0) - \frac{R_{\max}}{(1+\alpha)(1-\gamma)}\left(\frac{\ln(8/\tau)+|\pi^*|\ln\left(KD|A||Z|\right)}{\alpha K} + (1-\alpha)\left(\sqrt{\frac{2\ln(2/\tau)}{K}} + \gamma^D\right)\right). \tag{6}$$

**Proof.** By Theorem 1, with probability at least $1 - \tau/2$,

$$V_\pi \geq \frac{1-\alpha}{1+\alpha}\hat{V}_\pi - \frac{R_{\max}}{(1+\alpha)(1-\gamma)}\left[\frac{\ln(8/\tau)+|\pi|\ln(KD|A||Z|)}{\alpha K}\right].$$

Suppose the above inequality holds on a random set of $K$ scenarios. Note that there is a $\pi' \in \Pi_{b_0,D,K}$ which is a subtree of $\pi^\star$ and has the same trajectories on these scenarios up to depth $D$. By the choice of $\pi$ in Inequality (5), it follows that with probability at least $1 - \tau/2$,

$$V_\pi \geq \frac{1-\alpha}{1+\alpha}\hat{V}_{\pi'} - \frac{R_{\max}}{(1+\alpha)(1-\gamma)}\left[\frac{\ln(8/\tau)+|\pi'|\ln(KD|A||Z|)}{\alpha K}\right].$$

Note that $|\pi^\star| \geq |\pi'|$, and $\hat{V}_{\pi'} \geq \hat{V}_{\pi^\star} - \gamma^D R_{max}/(1-\gamma)$ since $\pi'$ and $\pi^\star$ only differ from depth $D$ onwards, under the chosen scenarios. It follows that with probability at least $1 - \tau/2$,

$$V_\pi \geq \frac{1-\alpha}{1+\alpha}\left(\hat{V}_{\pi^\star} - \gamma^D\frac{R_{max}}{1-\gamma}\right) - \frac{R_{\max}}{(1+\alpha)(1-\gamma)}\left[\frac{\ln(8/\tau)+|\pi^\star|\ln(KD|A||Z|)}{\alpha K}\right]. \tag{7}$$

By Lemma 3, with probability at least $1 - \tau/2$, we have

$$\hat{V}_{\pi^\star} \geq V_{\pi^\star} - \frac{R_{\max}}{1-\gamma}\sqrt{\frac{2\ln(2/\tau)}{K}}. \tag{8}$$

By the union bound, with probability at least $1 - \tau$, both Inequality (7) and Inequality (8) hold, which imply Inequality (6) holds. This completes the proof. □