[Reviews · NeurIPS 2013]

Submitted by Assigned_Reviewer_1

The paper presents a new online approach for solving POMDPs. The paper builds on some solid work, including POMCP, AEMS2, which are some of the state-of-the-art methods for this problem. The key new insight presented is to prune the forward search tree using regularization of the policy size. Another contributions is a performance bound on the value estimate; this is used in the algorithm to direct the search. The paper includes a number of empirical results, comparing with other recent POMDP methods (online and offline). The new LaserTag domain is also a good contribution, which would likely be re-used by other researchers.

The core idea of controlling the complexity of the online search using constraints on the policy size is an interesting one, which has not been exploited in online POMDP methods. It provides good insight into the key question of balancing computation and solution quality in POMDPs. This is probably the most interesting aspect of the paper, which could have a major impact on solving large POMDPs. The authors suggest they will share the code - this would definitely be a plus.

Even though I liked this idea, it's hard to predict what will be the impact of the work. In particular, looking at the empirical results, it's not clear to me what aspects of the proposed R-DESPOT framework have the most impact in practice. Is is the particle filtering? The DESPOT structure? Or the regularization itself? These aspects really need to be teased out carefully so we have a clear picture of what is going on.

I have some concerns about the experiments. Why not re-compute the AEMS2 results? The algorithm should be simple to implement, given a DESPOT implementation. The results for AEMS2 are now 5 years out of date, and seem very competitive with what you report for DESPOT/R-DESPOT, at least on small domains. In the case of the larger domains, why not try AEMS2 with particle filtering belief representation, to see whether it's the belief filtering or the value approximation that really matters? This is a really important/interesting question, and the paper would be much stronger if it provided good evidence on this. Also, can you speculate on why your POMCP results are poorer than those in [3]? Overall, a more detailed discussion (Sec. 6) would be very useful to understand the impact of the work.

The paper is for the most part well written, especially the early parts, including the intro, related work, and basic DESPOT framework. In the latter parts, there are a number of technical details that should be clarified.

- Be *very* clear about what are the differences between DESPOT and R-DESPOT. Is it only the PRUNE function (line 301)? Can you give pseudo-code for this function?

- Summarize the key differences between POMCP and DESPOT. I think both use the particle filtering representation of the belief. So it is mostly the use of the lower/upper bounds in DESPOT? This can be in Sec. 3 or Sec.6.

- How loose is the lower bound? Do you use the actual bound in expression (1), or just some hand-tuned parameter L?

- Add more detail on how the key parameters alpha and K (or L) are picked experimentally. Did you use the same ones for all the experiments? How much offline time was used to select them? I understand this is a side issue, but I need to understand the procedure and the amount of work required for this (at least order of magnitude).

- How do you impose the time constraint (1 sec / action) in practice? Do you ignore the parameters (e.g. K) if they exceed the time limit?
Summary: This could be a very strong paper. It makes an potentially very interesting contribution to POMDP solving. But there is a lack of technical detail, which probably makes the results difficult to reproduce. And there are some non-trivial limitations in the results and discussion as presented, which make it difficult to assess whether the work will have a major impact or not. I look forward to the authors' response.

Submitted by Assigned_Reviewer_7

This paper presents a determinized sparse partially observable tree for online POMDP planning. For that, it derives an ouput-sensitive performance bound, showing thus that the online proposed approach works "well" if a simple policy exists. The online approach itself searches in a sparsely sampled belief tree for a policy that optimally balances between the size of the policy and the accuracy on its value estimate obtained through sampling.
Experiments show that the proposed algorithm scales up better than existing online methods.

This work appears however incremental and unlikely to have much impact even though it is technically correct and well executed. Indeed from the originality point of view, it is just a technical attempt to have an algorithm which scales up better. Unfortunately this attempt presents some limitations

(1) The proposed regularization does not appear beneficial to the RockSample(n,k) even if this is mostly deterministic

(2) The initial lower bound as suggested by the authors does not satisfy the requirements for Theorem 1 to hold and we do not what's the importance of that in practice.

(3) The proposed algorithm relies on tree search and we know that this type of search is very sensitive to the branching factor (which is the number of observation) and this is not taken into consideration neither theoretically nor experimentally. Although the authors stated " Theorem 1 indicates that R-DESPOT may still perform well when the observation space is large". However "MAY STILL" remains (for me) too fuzzy and I think a formal and practical analysis in terms of observations is required here.


It is not clear what is represented in Table 1, if it is a "running time" then why we have some results as -6.03; -6.19, etc.

The right reference for AEMS is :
ROSS, Stéphane et CHAIB-DRAA, Brahim. AEMS: An Anytime Online Search Algorithm for Approximate Policy Refinement in Large POMDPs. In : IJCAI. 2007. p. 2592-2598.
For its part [2] is a general overview containing many comparisons on online POMDP approaches.

The authors say and repeat all the time "the algorithm works well if simple policy exists". What does this mean? What does ``simple policy" mean? what can we do if we do not have a simple policy? should we ask some expert?

Summary: An incremental online POMDP planning of which the branching factor is not taken into consideration and the experiments are not convincing.

Submitted by Assigned_Reviewer_8

---Response to Author Response---

Thank you for the clarifications. Just a couple of points:

- While I do agree that characterizing the class of problems that have small optimal policy trees would be interesting, I think the paper could do a better job of describing the approach's strengths and weaknesses, even without fully answering this theoretical question. The authors must surely have some intuitive examples of problems where this algorithm should do very well, and problems on which it should do poorly. Sharing this intuition would significantly strengthen the paper in my opinion.

- Relatedly, whether DESPOT is better at exploiting certain kinds of prior information was only part of my question. It is also important to address whether DESPOT *requires* prior knowledge to do well. From that perspective I don't think empirical comparison is difficult; you can simple try out DESPOT with an uninformed default policy (e.g. uniform random). If it does well, it would alleviate concern about the need for a good default policy. It if does poorly, it will be illustrative of problems for which DESPOT would not be a good choice. Either way it would increase the knowledge transmitted by the paper without requiring much additional effort or space.

---Summary of paper---

The paper presents an online POMDP planning algorithm that uses both classical search methods and Monte Carlo sampling to select an action. The main idea of the algorithm is to pre-sample random events (a la PEGASUS) in the roll-out tree, and then compute a good policy for the determinized future (values at the leaves are estimated using Monte Carlo samples of a default policy). The policy search is regularized to prefer "smaller policies" (policies that fall back on the default policy after a small number of steps) and the determinized tree can be pruned using a heuristic branch-and-bound algorithm to reduce the computational cost of finding a policy. Performance bounds are given in terms of the number of samples, depth of the tree, *and* the size of the optimal policy on the tree. Experiments show the algorithm performs well in several large POMDPs in comparison to contemporary competing methods.

---Quality---

As far as I can tell the analysis is technically correct. The bounds derived give insight into the algorithm's properties (though it's not clear how much purchase the bounds have once the DESPOT is approximated) and for the most part match the intuitions given.

The main thing I would have liked to see in this paper is more of a discussion of the strengths and weaknesses of this approach, particularly in comparison to the UCT-like approach (POMCP). The introduction claims that R-DESPOT has better worst-case behavior, but this is only the case if the optimal policy tree is "small." I think the paper would be much stronger if it contained more discussion of what the implications of a "small" policy are. In what types of problems is this likely to be the case and when is it catastrophically not the case? I am most concerned that this may depend heavily on the choice of default policy (because a "small" policy is one that mostly defers to the default policy), meaning that for this method to perform well, one must already have a high quality policy available. I think that trying to exploit the existence of a simple optimal policy is sensible, but it's not clear to me that this is a good way to think about simple policies -- it would help to have a better idea of what the assumption means in terms of the problems it does/does not encompass.

Relatedly, and perhaps more importantly, it's not clear to me that the experiments are really a fair comparison. R-DESPOT is given a domain-specific default policy in each experiment. While I can tell that the authors have tried to keep these policies intuitive and simple to specify, they nevertheless seem to constitute prior knowledge that is not supplied to the other methods. It is hard, therefore, to tell how much of the performance gap is due to the method, and how much is due to the choice of default policy. I don't necessarily think this fatally undermines the results. I think it would be reasonable, for instance, to claim that R-DESPOT can exploit prior knowledge that other methods can't, or that would be a challenge to encode for other methods. If that is the claim, however, it needs to be stated explicitly and justified at least in intuition. Even if it is difficult to supply comparable prior knowledge to the other methods, I would have at least liked to have seen the performance of R-DESPOT with a uniform random default policy, even if it performed poorly, since this would be informative about whether R-DESPOT is appropriate for a problem in which I do not have an informed default policy.

---Clarity---

I found this paper to be clear and very readable. I also found the mathematics in the supplementary materials to be refreshingly well-presented and easy to follow.

A small suggestion: at first I found the paragraph at the end of Section 3 to be a bit out of place. Once in Section 4 I quickly realized that it was meant to foreshadow the R-DESPOT algorithm. This could be clarified with a simple "We will make use of this idea in the next section in order to develop a more practical algorithm" or similar.

Type-o near bottom of p. 1: behavoior

---Originality---

Though the method presented here makes use of several existing ideas, it combines them in a novel and interesting way. I felt the method was well-contextualized in the relevant literature.

---Significance---

The problem of planning efficiently in high-dimensional partially observable worlds is both important and challenging. The algorithm presented here is rooted in theoretical performance bounds and performs well empirically. Though I do still have questions regarding the need for prior knowledge, in some domains prior knowledge is available. The main potential for impact that I see is that the algorithm represents a seemingly viable alternative approach to UCT-like algorithms (as well as a different style of analysis) that may serve as inspiration for further algorithmic developments in this area.
Summary: The paper is clear; the problem is important; and the method is new, interesting, and reasonably well evaluated. Therefore I recommend the paper be accepted. That said, I think the paper could be made significantly stronger by including more intuitive discussion of the algorithm's strengths and weaknesses. Most importantly, I hope the authors will revise the experiment section to include explicit discussion of the role of prior knowledge in performance differences observed.
Author Feedback

Author rebuttal: We thank all the reviewers, especially reviewers 1 and 8 for their many constructive criticisms.

We must, however, disagree with reviewer 7 on most of his/her main criticisms:

* "An incremental online POMDP planning of which the branching factor is not taken into consideration and the experiments are not convincing." (reviewer 7)

"The proposed algorithm relies on tree search and we know that this type of search is very sensitive to the branching factor (which is the number of observation) and this is not taken into consideration neither theoretically nor experimentally." (reviewer 7)

The main criticism of the review summary is factually incorrect. In fact, one main idea of our algorithm is to construct search trees with small branching factor (DESPOT) and yet achieving good performance when |Z| is large. A DESPOT of height D has size bounded by KD|A|^D, where K is the number of random number sequences and |A| is the number of actions. In contrast, the normal belief tree of height D has size |A|^D |Z|^D, where |Z| is the number of observations. In addition, Theorem 1 shows the effect of observation space size on performance bound is small (ln |Z|). LaserTag provides empirical evidence that using a DESPOT with small K can handle millions of observations.

* "... just a technical attempt to have an algorithm which scales up better" (reviewer 7)
This statement contradicts the well-recognized consensus of the area: scaling up is one of the most important challenges for POMDP planning. It is also inconsistent with the impact assessment of the other two reviewers.

* "The proposed regularization does not appear beneficial to the RockSample(n,k) even if this is mostly deterministic"

The paper states so. Indeed, less randomness would mean less overfitting and less regularization is needed.

We thank reviewer 1 for the suggestion of comparing with AEMS2 with particle belief representation. Although we couldn't do it now due to time and page limitation, we will definitely try it.

Below we summarize our response to reviewer 1 and 8's main questions over DESPOT and its comparison with AEMS2 and POMCP.

Regularized DESPOT has 3 main technical ideas:
1. Use simulation traces instead of belief updates in the search tree;
2. Use pre-sampled random sequences to dramatically reduce the branching factor of the DESPOT tree, in particular, observation branches, while guaranteeing the quality of approximation (Theorem 1)
3. Use regularization to bias towards simple policies, and avoid overfitting to pre-sampled random sequences

Both (1) and (2) are essential for the algorithm to scale up. (3) holds under the condition that a simple policy exists.

In comparison, AEMS2 has neither (1) nor (2). Even with particle belief representation, it may still have difficulty scaling up. Particle filtering is computationally expensive, if it is used in every node of the search tree. In contrast, DESPOT does not use particle filtering during search, but only do belief update when a real observation is received. Further, for AEMS2, the search tree of height D has size |A|^D |Z|^D, which is impractical for large observation spaces, such as in LaserTag.

POMCP and DESPOT share (1), but the search strategies of DESPOT and POMCP are very different. POMCP can be badly misled by its UCB search heuristic. The worst-case bound for POMCP's run time is a tower of D-1 exponentials. The worst-case run time for DESPOT is bounded by the DESPOT tree size, KD|A|^D. Theorem 1 shows that the number of sequences K required grows roughly linearly with the policy size (|Z|^D in the worst case). Plugging in to the DESPOT size of KD|A|^D, we get D|A|^{D}|Z|^D, giving us a worst-case run time that is exponential in D, but not POMCP's tower of D-1 exponentials.


Detailed responses to specific questions:

Reviewer 1
* Difference between ours and [3]'s POMCP results on RockSample
The original POMCP implementation ([3]) does not guarantee that online action selection occurs within 1 sec, but achieves so on the average from a posteriori time measurement. We modified the code to guarantee the 1 sec constraint for both DESPOT and POMCP. The machines used in [3] may also be much more powerful than ours. We actually tried running POMCP much longer (16 times longer), and get roughly the same performance as [3].

* Relationship between DESPOT and R-DESPOT
DESPOT is R-DESPOT without regularization (the prune function is provided in the supplement).

* Parameter tuning
We fixed K=500 in our experiments. We tuned the regularization constant L empirically by offline simulation using a training set distinct from online test set, using a few hours with 1 sec per action.

* Time limit for online action selection
We measured the time consumed so far after each trial, and terminated the search when the time limit is exceeded.

Reviewer 7
* "The initial lower bound as suggested by the authors does not satisfy the requirements for Theorem 1 to hold and we do not what's the importance of that in practice."
The paper states this clearly. Our results suggest empirically this gives a good approximation, though more investigation is needed for better understanding.

* Table 1 reports average returns.

* We will add the suggested reference for AEMS2.

* A simple policy means a policy with small size, measured in the number of belief nodes. Theorem 2 shows that DESPOT does better when a small optimal policy exists.

Reviewer_8
* Effect of optimal policy size
We fully agree that characterizing the class of problems with small optimal policies is an interesting and important issue, and have put it on our to-do lis.

* Influence of default policies in experimental results
DESPOT can indeed better exploit prior knowledge. This makes experimental comparison with other methods more difficult. We will clarify this issue.